# Analyzing the Spatial Equity of Walking-Based Chronic Disease Pharmacies: A Case Study in Wuhan, China

**DOI:** 10.3390/ijerph20010278

**Published:** 2022-12-24

**Authors:** Yue Liu, Yuwei Su, Xiaoyu Li

**Affiliations:** 1School of Natural and Built Environment, Queen’s University Belfast, Belfast BT9 5AG, UK; 2School of Urban Design, Wuhan University, Wuhan 430072, China; 3Department of Landscape Architecture & Urban Planning, Texas A&M University, College Station, TX 77843, USA

**Keywords:** spatial equity, primary healthcare, walking accessibility, 15 min city

## Abstract

Chronic diseases place a substantial financial burden on both the patient and the state. As chronic diseases become increasingly prevalent with urbanization and aging, primary chronic disease pharmacies should be planned to ensure that patients receive an equitable distribution of resources. Here, the spatial equity of chronic disease pharmacies is investigated. In this study, planning radiuses and Web mapping are used to assess the walkability and accessibility of planned chronic disease pharmacies; Lorenz curves are used to evaluate the match between the service area of the pharmacies and population; location quotients are used to identify the spatial differences of the allocation of chronic disease pharmacies based on residents. Results show that chronic disease pharmacies have a planned service coverage of 38.09%, an overlap rate of 58.34%, and actual service coverage of 28.05% in Wuhan. Specifically, chronic disease pharmacies are spatially dispersed inconsistently with the population, especially the elderly. The allocation of chronic disease pharmacies is directly related to the standard of patients’ livelihood. Despite this, urban development does not adequately address this group’s equity in access to medication. Based on a case study in Wuhan, China, this study aims to fill this gap by investigating the spatial equity of chronic disease medication purchases.

## 1. Introduction

The term chronic disease refers to chronic non-communicable diseases. They have an adverse impact on the economic development of a nation. With global urbanization, the prevalence of chronic diseases increases [1]. According to the academics of WHO, chronic diseases will cause around 69% of deaths by 2030 [2]. For China, chronic diseases have become a major public health problem at the national level [3]. China’s State Council issued the Medium and Long-term Plan for the Prevention and Treatment of Chronic Diseases (2017–2025) and the Health China Action (2019–2030), which emphasize the prevention and treatment of chronic diseases.

One of the effective ways to promote public health is to promote equity in health and social services in urban governance [4]. Echoing this, China is in the process of reforming its primary healthcare system combined with walking-based 15 min life units to ensure equitable access to public services and promote full coverage of health services. On this, Wuhan is building 15 min walking community circles with health service [5]. This kind of implementation is derived from the concept of ‘15 min city’ which emphasizes the proximity and accessibility of facilities and services around residences [6].

Current urban research studies in China are concerned with public healthcare accessibility for the construction of the 15 min city while failing to consider the joining of non-public facilities, such as retail pharmacies. Furthermore, previous relevant studies have not adequately considered the daily medicine shopping needs of patients with chronic diseases. This paper aims to fill in these gaps. The specific purposes are to: (1) propose a method for measuring spatial equity in primary care facilities; (2) identify spatial inequalities in chronic disease pharmacies in 15 min walkable neighborhoods; and (3) propose optimization strategies for planning primary chronic disease pharmacies in the main urban area of Wuhan.

## 2. Literature Review

### 2.1. Study on Primary Healthcare

Primary healthcare facilities bring healthcare services close to families and communities [7]. Traditional primary healthcare services are vital and necessary, although telehealth is shaping a new healthcare system. Telehealth covers telemedicine, telenursing and telepharmacy, which can extend the reach of clinical interventions, increasing patient access to healthcare [8,9]. However, telehealth practices indicate that some limitations exist, such as reimbursement of telemedicine services, limited in-person physical exams, low accessibility for older adults and lower socioeconomic classes to internet services and smartphones, etc. [10,11,12,13]. Thus, there is widespread research on the accessibility of primary healthcare services [14,15,16,17,18]. Existing studies discussing the accessibility of primary healthcare facilities focus on the needs of the vulnerable, such as barrier-free travel for the elderly and walking for low-income families [14,17]. The need for a more refined scale analysis (at the community level) has also been suggested [15,19].

However, there is a gap in the types of primary health service facilities in these studies, especially in China. The current scholarly focus on China’s primary healthcare system is public facilities while ignoring the complementary non-public facilities, such as retail pharmacies. This condition is mainly because of the specialty of China’s health system: Prior to the year 2000, China’s healthcare delivery was dominated by public hospitals, while the construction of primary healthcare facilities was not given priority [20]. These hospitals contain outpatient pharmacies, which provide the primary source of medication for patients. With the new round of health reform in the late 2000s, the government began to emphasize the primary healthcare’s significance and encouraged non-public forces to participate in the provision of medical services [15].

In addition, although some scholars have begun to study primary healthcare regarding the 15 min city in China, attention needs to be paid to the needs of different patients [6,21]. Especially for patients with chronic diseases, health insurance discounts are only available in specific primary outpatient and retail pharmacies [22,23]. Chronic diseases are a collective term for a wide range of diseases. The four most common chronic diseases are cardiovascular diseases, diabetes, malignant neoplasms, and chronic respiratory diseases [24]. Most of these disorders are challenging to cure and often do not require long-term hospitalization but daily medication due to their insidious onset, long duration, and slow progression. Adherence to medical therapy has become a necessity for patients with chronic diseases to promote illness control [25,26]. Therefore, it is necessary to understand the universality of special primary healthcare facilities, i.e., chronic disease pharmacies.

### 2.2. Study on Spatial Equity

Spatial equity emphasizes the differences in the facilities enjoyed by different areas or social groups [27]. As regards urban facilities planning, spatial equity refers to the relationships between residents and facilities, i.e., spatial equality or spatial proximity [28]. This spatial equity is, to some extent, an extension of the concept of accessibility [29]. Therefore, spatial accessibility evaluation is widely used to evaluate the spatial equity of public service facilities [14,15,16,17,18,30]. Spatial accessibility refers to the ease with which different spatial units can overcome the costs (time, distance, etc.) involved in reaching the desired facility. One of the critical factors affecting spatial equity is spatial distribution characteristics [31]. Assessing spatial accessibility can effectively identify areas where public services are scarce, thus providing a quantitative basis for spatial equity research [32,33].

In terms of methods, the development of GIS technology in the 1990s led to the ArcGIS being favored by different academics, with common ones being buffer analysis, potential models, and gravity models [15,18,34]. After the 2000, along with the continuous maturation of Internet technology, traditional ArcGIS analysis has also been gradually linked to ‘big data’ and Web mapping services. Online platforms, such as Google Maps and Yahoo Maps, provide publicly available data. The application programming interface (API) allows access to real-time optimal route choices under different travel modes, corresponding time cost information, social data, media data, user comment data, etc. This feature of Web mapping ensures the validity, convenience and accuracy of the accessibility metric results [35,36,37].

In fact, there are still limitations to assessing spatial equity if only considering accessibility alone. Thus, actual spatial equity evaluations are not only based on accessibility, but also complemented by consideration of the needs of different social groups. Commonly used measures of spatial equity include correlation analysis, the Gini coefficient, the Lorenz curve, and locational quotients [15,38,39,40].

Following the review of primary healthcare facilities and relevant methods, this research set a workflow to evaluate the equity of medicine purchase for patients with chronic diseases: the main urban area of Wuhan, China, was selected as the study area; Web mapping was used to measure the walkability to pharmacies; Lorenz curves were used to evaluate the match between pharmacies’ service coverage and population; and locational quotients were used to identify differences in the spatial distribution of pharmacies’ coverage on populations.

## 3. Methodology

### 3.1. Study Area

According to China’s Sixth Health Services Statistics Report 2018, the central region has the highest prevalence of chronic diseases in urban areas, at 34.7% [41]. Wuhan is the center of development in central China, a mega-city in the Yangtze River Economic Belt, and a major national strategic development area in China. As of 2021, Wuhan has a population of 13,648,900 and 1,456,200 residents over the age of 65 [42,43]. About 32.47% of Wuhan’s population suffers from chronic diseases [44].

The setting of this study is the main urban area of Wuhan (Figure 1), including Jiangan District, Jianghan District, Qiaokou District, Hanyang District, Wuchang District, Hongshan District and Qingshan District (including the East Lake Scenic Area, Wuhan Economic and Technological Development Zone and East Lake High-Tech Development Zone), with a total study area of about 678 km^2^.

### 3.2. Data Preparation

#### 3.2.1. Pharmacy Data

According to the Spatial Planning Guidance to Community Life Unit in China, the service radius of a 15 min life unit is 1 km, so this study extracted chronic disease pharmacies from the main urban area and its 1 km buffer zone [45]. A total of 254 chronic disease pharmacies (as of the early 2019) were identified based on the public announcement list of health and wellness bureaus in each district, along with official media websites such as Changjiang Daily, Jingchu.com (Figure 2). The physical addresses of pharmacies are converted from to geographic coordinates through GeoSharp (Version 2.0, https://www.udparty.com (accessed on 22 December 2020)).

#### 3.2.2. Population Data

The population data is the 2017 Wuhan city street-level population data provided by the Wuhan Land Resources and Planning Information Centre.

#### 3.2.3. Road Data

The road data was obtained from Gaode Map (https://ditu.amap.com (accessed on 23 January 2019)). The road data also requires to be fixed line errors by geodatabase topology in ArcGIS (Version 10.3, http://www.esri.com (accessed on 1 January 2019)).

### 3.3. Research Method

#### 3.3.1. The Framework for Chronic Disease Pharmacies Equity Measurement

The method framework is shown in Figure 3. First, in terms of geospatial service coverage, ArcGIS 10.3 combined with Web mapping is used to measure the planned service area and walking accessibility of chronic disease pharmacies. A comparison of the two coverages interrogates the gap between planned and actual service coverage. Second, regarding the allocation of service coverage, the Lorentz curve is used to understand the match between the pharmacies’ service coverage and the population. Finally, the location quotients are calculated to identify differences in the allocation of service coverage among regions.

#### 3.3.2. Measuring Planned Service Coverage by Buffer Model

The service radius of outpatient pharmacies is defined as 1000 m and 500 m, respectively, based on the settings defined in the Spatial Planning Guidance to Community Life Unit [45]. In the case of retail chronic disease pharmacies, there is no relevant planning guidance. As they are oriented to serve the surrounding communities, the walking range radius corresponding to a 15 min life unit is set at 1000 m. The buffers were constructed in ArcGIS 10.3.

Service coverage indicates the service area covered by the pharmacies within each district as a proportion of their district’s area. If a pharmacy’s service area covers external jurisdictions, the external service area will be calculated in the external jurisdictions, but the overlapped coverage will not be calculated. The service overlap rate is the ratio of the overlapped service area within a jurisdiction to the sum of the service areas of all pharmacies in that jurisdiction (effective service area). The lower the overlap rate, the less duplication and waste of resources.
(1)$$C_{i}=\frac{\sum {\mathrm{PA}}_{i}}{A_{i}}$$
(2)$$O_{i}=\frac{\sum {\mathrm{CO}}_{i}-\sum {\mathrm{PA}}_{i}}{\sum {\mathrm{CO}}_{i}}$$
where C*_i_* denotes the service coverage of chronic disease pharmacies in jurisdiction *i*; O*_i_* denotes the service overlap rate of chronic disease pharmacies in jurisdiction *i*; PA*_i_* denotes the service area of the vertical projection of chronic disease pharmacies in the region; CO*_i_* denotes the service area of pharmacies in jurisdiction *i*, i.e., the effective service area, and A*_i_* denotes the size of jurisdiction *i*.

#### 3.3.3. Measuring Walking Time by Web Mapping

The process is shown in Figure 4. Firstly, the fishnets are constructed using ArcGIS 10.3. To ensure accuracy, this study applies 100 m × 100 m uniform point arrays within the main urban area of Wuhan based on the walking distance of a 15 min life unit. Next, with Python, the walking navigation API of Gaode Map is used to calculate the walking time between each fishnet and the three nearest chronic disease pharmacies. Additionally, then, this data is sorted and aggregated by Excel to filter out the shortest time from each fishnet to the nearest pharmacy (The essence of calculating accessibility using Web mapping is to solve for the time it takes for each point of fishnet to reach each facility. Solving directly for the optimal route solution means N1 × N2 calculations (N1 is the number of fishnets and N2 is the number of pharmacies), which is beyond the availability of map suppliers and the computing power of computers. In addition, most fishnets are not linked to pharmacies as they span multiple areas between them. Hence, to reduce the amount of data calculated online, the data needs to be pre-processed offline to reduce the number of calculations to N1 × 3 times). Finally, the fishnets are assigned attributes by jurisdiction. The number of fishnets in each jurisdiction is counted by filtering by attribute field and time interval. The actual coverage based on walking time is the fishnets within the 15 min life units as a proportion of the total number of fishnets in the area. Multi-temporal data extraction is not conducted because this research only examines walking medicine purchase behavior and real-time traffic conditions have little influence on the findings.

#### 3.3.4. Measuring Equality of Service on Chronic Diseases Pharmacies

Equity is assessed on two levels, namely, per-capita and inter-regional. In this study, the service buffer of chronic disease pharmacies is used to represent the service resources. By using ArcGIS extraction analysis, a proportion of the study area is calculated for each street’s effective service area. The proportions are ranked from lowest to highest. Then, the cumulative percentage of the area served, as well as the resident population and the elderly population, are plotted to form the Lorenz curve and their fitted curves.

The Lorenz curve does not reflect the spatial match between the distribution of pharmacies and the urban population. Therefore, this research applies location quotients to measure the match between the service coverage of chronic disease pharmacies and the population distribution. The calculation formula is as follows.
LQ*_i_* = (M*_i_*/P*_i_*)/(M/P)(3)
where, LQ*_i_* is the location quotient of chronic disease pharmacies’ service in the street *i*; M*_i_* is the effective service area of chronic disease pharmacies in the street *i*; P*_i_* is the number of resident populations in the street *i*; M is the sum of effective service area of chronic disease pharmacies in the study area; and P is the resident population in the study area.

When LQ*_i_* > 1, it means the per capita access to the resources in the street *i* is higher than the overall level. Additionally, there is a high probability that the facilities’ resources match or are more adequate than the street population; When LQ*_i_* < 1, it indicates the per capita access to resources is lower than the overall level and that there may be a lack of resources [39].

## 4. Results

### 4.1. Accessibility by the Planned Service Coverage

Figure 5 shows that the service coverage of chronic disease pharmacies in the study area is 38.09%, while the service overlap rate is 58.34%, with the latter being 20.25% higher than the former. In the south of the Yangtze River, the service coverage of chronic disease pharmacies is lower than that north of the Yangtze River. However, the south’s overlap rate is lower, with an average overlap rate of 43.04% for the south and 57.21% for the north.

### 4.2. Accessibility by Walking

The coverage of services within a 30 min walk is 62.52% (Figure 6). The service area within 15 min walking time accounts for 28.05% of the district area, which is about half the size of the 15 min life units in the main urban area of Wuhan (53.94% of the district area) [46].

The actual service coverage based on 15 min walking time decreased by 10.04% compared to the planned service coverage (Figure 7). The change in coverage in Qingshan District and Jianghan District is below 6% (the former 5.12% and the latter 5.67%). The changes in Wuchang, Jiangan and Hanyang districts are at an intermediate level, at 9.74%, 7.64% and 7.44%, respectively. Hongshan District and Qiaokou District show more significant changes in coverage, with the former shrinking by 11.13% and the latter by 11.66%.

### 4.3. Equality of Service on Chronic Diseases Pharmacies

#### 4.3.1. Per Capita Equity by the Lorenz Curves

The Lorenz curves for chronic disease pharmacy services by population are depicted in Figure 8. The fitted curves constructed correspond to R-squared values of 0.9986 and 0.9984, respectively, indicating that they are close to the original curves.

According to the fitted curves, 60% of residents shared 47% of the pharmacies’ services, while the remaining 40% had 6% more resources than the former. The unevenness is much more evident in the calculations for the elderly. 70% of the old only shared 52% of the pharmacies’ services, while the remaining 30% had almost as many resources as the former.

#### 4.3.2. Inter-Regional Equity by Location Quotients

The location quotients are classified into five classes using natural discontinuity analysis (Table 1 and Table 2). In terms of location quotients based on resident population (Figure 9a), 40.86% of streets in the study area are below the medium level of chronic disease pharmacies’ services per capita; 11.83% of streets are in the extremely high or higher bracket; residents within streets in the exceptionally high bracket enjoy more than three times the overall average level of service area per capita.

The results of the location quotients based on the elderly population show that 52.69% of the streets are below the overall level (Figure 9b). There are extremely low values observed at Minquan Street in Jianghan District (0.243) and Hanzhong Street in Qiaokou District (0.355). Conversely, areas with extremely high values include Hongshan Street in Hongshan District (14.267) and Chaoyang Street in Caidian District (8.244).

## 5. Discussion

This study simulated the currently planned service area of chronic disease pharmacies using buffers and measured the actual service area using Web mapping. It found a gap between the planned and actual use of chronic disease pharmacies. The overall coverage of the pharmacy service area is low, while the overlap is high, indicating the effective utilization of pharmacy resources is low. The accessibility of pharmacies decreases from the old city center to the periphery, and there is a spatial divergence between the north and south. This result supports the argument that old central districts provide higher pedestrian accessibility to public services [47,48].

The results of the Lorenz curve and location quotients reveal a gap in the current per capita service level of chronic disease pharmacies in Wuhan’s main urban areas. The distributional inequality is observed between groups with fewer service resources and those with more, and this inequality is particularly evident among the elderly. Recent research similarly indicates that there are pharmaceutical deprivation areas for the elderly, despite the illusion of good coverage implied at the metropolitan level [19].

### 5.1. Influencing Factors of Chronic Disease Pharmacies Equality

Three main factors contribute to differences in equity: the time of build-up of the area; the targets of the regional development; and the topography.

(1)There are differences in the layout and type of residential areas and the demographic structure between the old and new urban areas. The main built environment of Wuhan is along the Yangtze River and gradually expanded outwards. The layout of the city is divided by the traffic ring roads and the Yangtze River. The first and second rings are dominated by old communities with dense housing and roads. Between the second and third rings, there are mainly old residential areas with high floor area ratios, low building density, and massive commercial areas due to urban renewal. Hence, walking accessibility is generally better within the second ring. The third ring and beyond are mainly new or under-construction high-end residential areas with low occupancy rates and poorly configured facilities. The clusters of residential areas are mostly separated by wide roads.(2)The different development targets of the seven jurisdictions have resulted in different plans for the built environment and public facilities. For example, Jianghan District has the highest economic power and ranks first in service coverage, while Hanyang District and Hongshan District are at the end of the list due to their late urban and economic development. In the Qiaokou District, Yijia Street and Gutian Street are mainly used for commercial services and public facilities, with many industrial parks and scattered residential land. Those areas are divided by railway lines and many urban highways with a width of 15 m or more, resulting in poor accessibility to the pharmacies.(3)Wuhan’s many lakes and hilly terrain have two impacts on the layout of pharmacies: firstly, the early construction of the city was limited by the lakes. Many of the new residential areas came about after the transport road network had been improved. Thus, most of those close to the third ring are upscale areas with an enormous footprint, and the construction of primary medical institutions has not yet been kept up. Secondly, the existence of an extensive lake system leads to road bypasses. Therefore, the layout of facilities is more dispersed, and the continuity of services is affected.

### 5.2. Optimizing Strategies for Chronic Disease Pharmacies Equity

This study proposes optimization options that can be linked to three perspectives: demographic structure; adjustment of the pharmacies’ layout; the form of pharmacy services.

(1)With limited facility resources, priority should be given to elderly patients with lower location quotients. The leading group of chronic disease patients is the elderly, with limited mobility. Each district government should grasp the changes in the spatial distribution of chronic disease patients, especially the elderly. Governments should adjust the pharmacy locations according to the space of the life unit, traffic structure, patient population changes, and trends to plan.(2)The current public chronic disease pharmacy system in Wuhan has already been established. So, it is more feasible to adjust the layout of existing facilities than to re-establish a dispensing facility system. In terms of investment, pharmaceutical companies invest in retail pharmacies and do not require financial investment from the government. There is no lack of outpatient pharmacies on each street within the second ring. Hence, retail pharmacies are encouraged to be introduced in blind areas, effectively reducing the government’s financial burden. Furthermore, some areas have a dense distribution of pharmacies and significant resource redundancy, requiring a moderate replacement of chronic disease pharmacies. Public outpatient pharmacies should be considered first for areas with insufficient chronic disease pharmacies.(3)Finally, optimizing facilities and services should also be linked to the Internet. Although most chronic disease medicines purchased online are not currently reimbursed in China, the future health insurance system will certainly interact more with the Internet, which will further enhance the resilience of cities in the future. It will play a pivotal role in the urban system, especially in response to public health emergencies such as the COVID-19 epidemic. The online purchase and reimbursement of medicines will lead to a new layout of primary healthcare facilities.

## 6. Conclusions

This research analyses the spatial pattern of chronic disease pharmacy resources based on space and population, using the central city of Wuhan as a case site. The results show that Web mapping services can estimate the accessibility of facilities on a finer and more accurate scale. The approaches in this study can further examine the equity of service provision of public service facilities under planning in conjunction with population data. The timing of urban build-up, development targets, and topography determine the distribution and structure of the population and road networks. The city’s distribution and structure of the population, as well as the road network, influence the accessibility and equity of pharmacy services.

In conclusion, the main contribution of this study is to propose a workflow for evaluating the equity of the layout of chronic disease pharmacies based on the Web mapping API, Lorenz curves, and location quotients. It focuses on the needs of chronic disease patients and complements the study of Web mapping for evaluating community life unit facilities. This workflow is based on the available dataset and has the potential to be applied in other regions or facilities, considering a lot of countries and regions have accessible demographic, household, and point of interest (POI) data. For example, it could be used to explain the potential inequality of other facilities’ locations in the 15 min city or life units, such as large grocery stores, parks, and recreational facilities [21]. However, the workflow cannot be simply applied to other regions without considering the various travel modes, such as in the U.S., where people made 89% of their trips by automobile and only 8% on foot [49]. For a region that heavily depends on vehicles, this framework is limited by ignoring real-time traffic data. A future study might focus on generalizing the model by including more types of travel modes. Meanwhile, considering that real-time traffic data are instantaneous, future studies might involve more informatics to achieve automatic or semi-automated results. On this, the automatic workflow could be combined with IoT collected-clinical data and be an effective tool in emergency medicine [37,50].

### Limitation

There are still some limitations in this study, which are areas for further improvement. In terms of data acquisition, the data on the distribution of the population with chronic diseases is not publicly available and can only be estimated by the number of resident populations in each street. In terms of walking accessibility calculations, Web mapping fails to measure the walking speed of the elderly. Future research into the accessibility of facilities for patients of different ages will be more relevant in guiding the planning and layout of pharmacies for chronic diseases.

## Figures and Tables

**Figure 1 ijerph-20-00278-f001:**
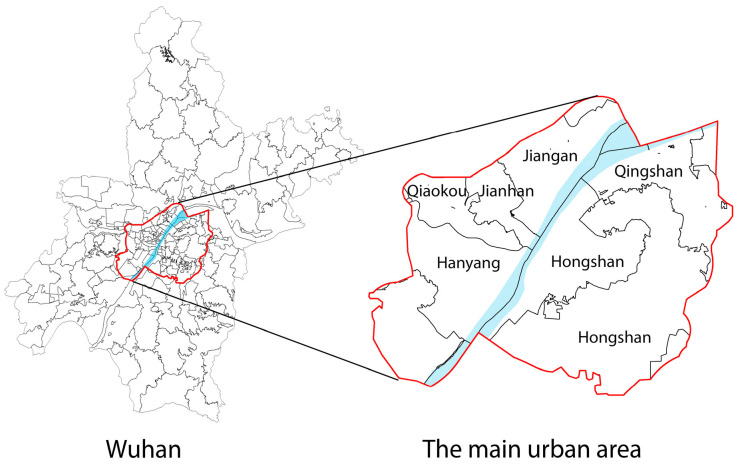
Location of the study area in Wuhan.

**Figure 2 ijerph-20-00278-f002:**
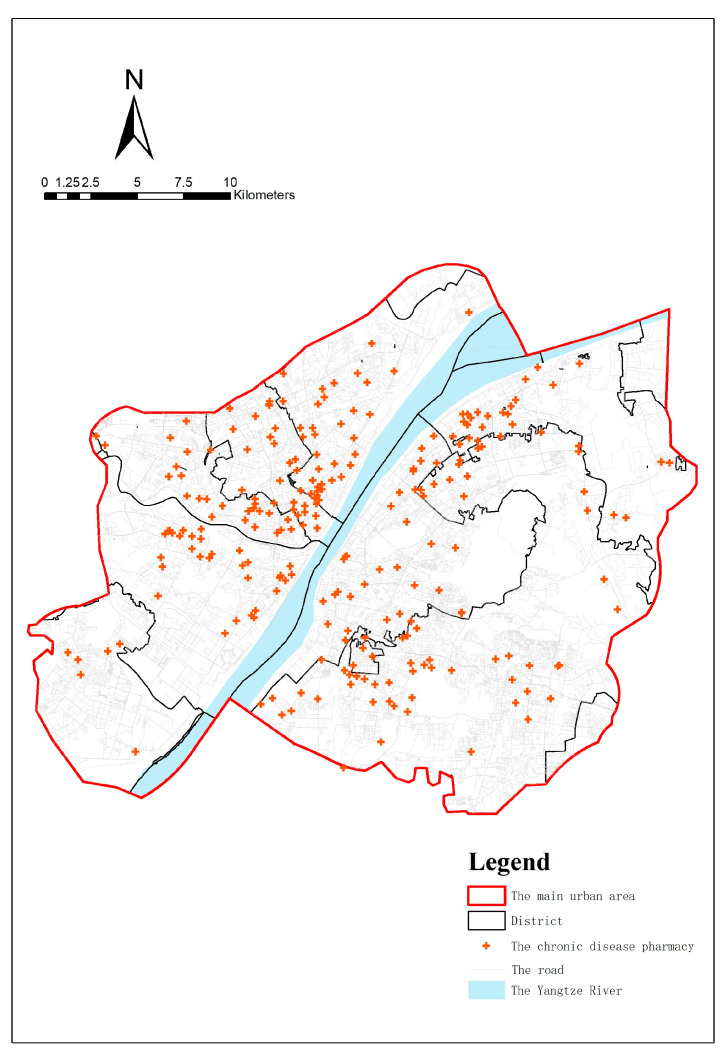
Distribution of chronic disease pharmacies and traffic road network in the main urban area of Wuhan.

**Figure 3 ijerph-20-00278-f003:**
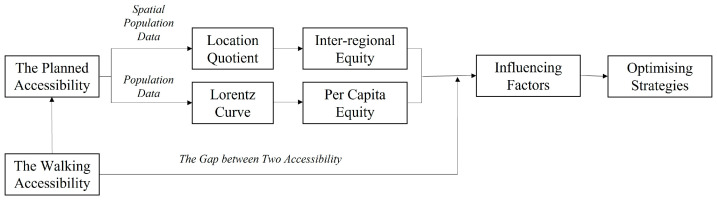
Methods workflow.

**Figure 4 ijerph-20-00278-f004:**
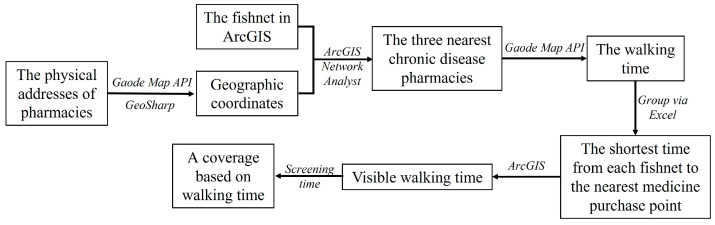
The workflow of measuring walking time by Web mapping.

**Figure 5 ijerph-20-00278-f005:**
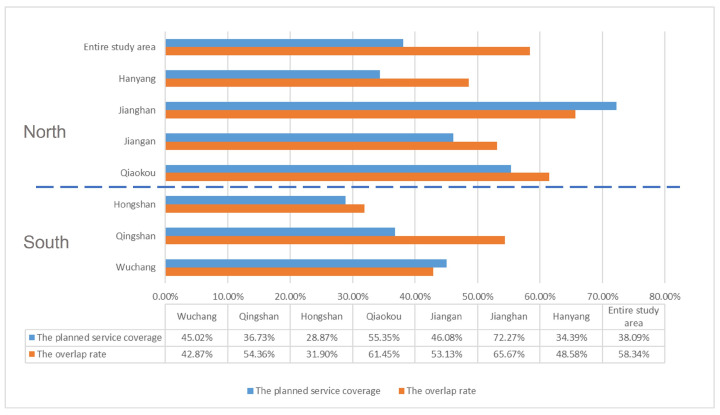
Chronic disease pharmacy service coverage and overlap based on planning radius.

**Figure 6 ijerph-20-00278-f006:**
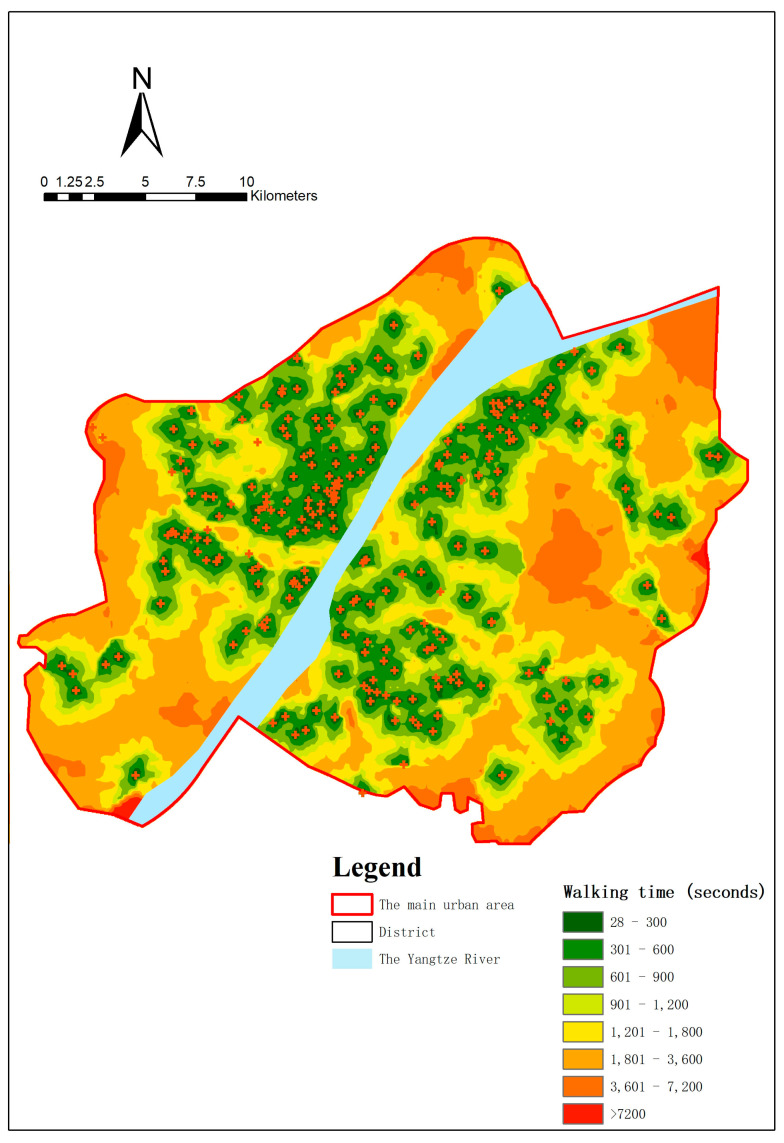
Actual service areas by walking.

**Figure 7 ijerph-20-00278-f007:**
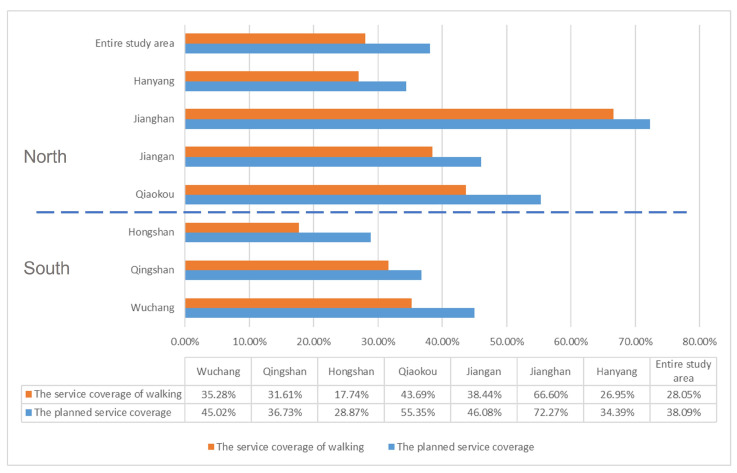
Planned service areas for chronic disease pharmacies vs. actual service areas based on walking.

**Figure 8 ijerph-20-00278-f008:**
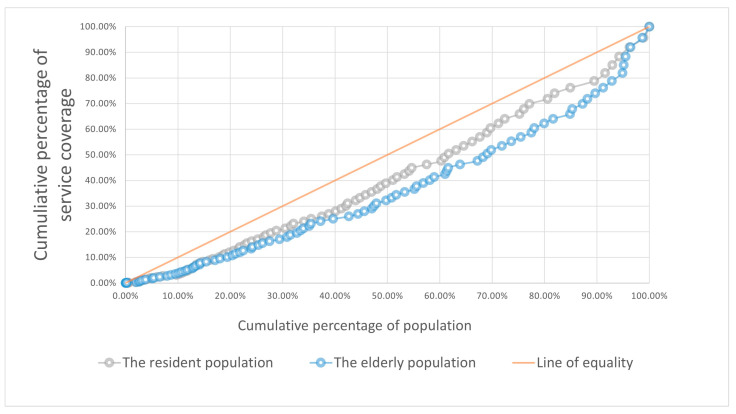
Lorenz curve by resident/elderly population.

**Figure 9 ijerph-20-00278-f009:**
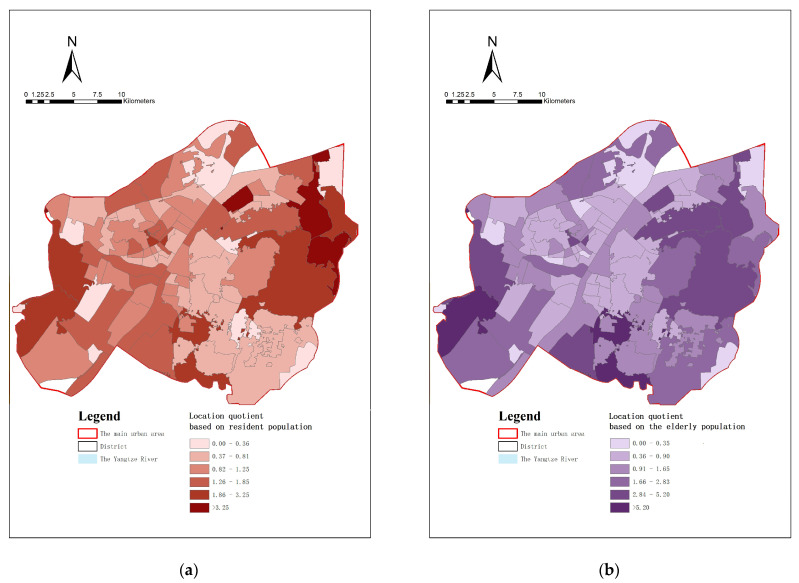
(**a**) The location quotient based on resident population; (**b**) The location quotient based on the elderly population.

**Table 1 ijerph-20-00278-t001:** Number and proportion of streets graded based on location quotient of the resident population.

Location Quotient Rating	Location Quotient	The Number of Streets	Percentage (%)
Extremely low	≤0.37	11	11.83
Relatively low	0.37–0.82	27	29.03
Medium	0.82–1.26	26	27.96
High	1.26–1.85	18	19.35
Extremely high	1.86–3.25	6	6.45
Exceptionally high	>3.25	5	5.38

**Table 2 ijerph-20-00278-t002:** Number and proportion of streets graded based on location quotient of the elderly population.

Location Quotient Rating	Location Quotient	The Number of Streets	Percentage (%)
Extremely low	≤0.36	10	10.75
Relatively low	0.36–0.91	39	41.94
Medium	0.91–1.66	24	25.81
High	1.66–2.84	8	8.60
Extremely high	2.84–5.20	9	9.68
Exceptionally high	>5.20	3	3.23

## Data Availability

Not applicable.

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
