# Peer review of "Analyzing the Spatial Equity of Walking-Based Chronic Disease Pharmacies: A Case Study in Wuhan, China"

_ijerph, 2022, doi:10.3390/ijerph20010278_

Round 1

Reviewer 1 Report

I have only one minor comment:

The authors are advised to consider mention telemedicine/telehealth/telepharmacy advantages and disadvantages

Reviewer 2 Report

From the computer science point of view, the paper lacks originality, but reading the aims and scope of the journal, it seems that this kind of studies are welcomed.

In the light of this, the study can be considered useful for public health purposes, but I think it can be improved by better surveying the information technologies that can be used for the study, and discussing how such results can be extracted in an automatic or semiautomatic way, also taking into account possible evolutions of underlying data. Withough an overlook on automation such a study lacks utility.

I would also suggest to enlarge this kind of study to the availability of emergency rooms for life-critical emergencies, like earth attack or sepsy, which might need medical intervention in a short time, like discussed in the following article:

L. Caruccio, et al., “Secure IoT analytics for fast deterioration detection in emergency rooms”, IEEE Access, Vol. 8, 2020.

Reviewer 3 Report

General Comments:

The subject addressed in this article, "Analyzing the Spatial Equity of Walking-Based Chronic Disease Pharmacies: A Case Study in Wuhan." is worthy of investigation. The paper presents interesting results about the current spatial distribution and locations of pharmacies in Wuhan, China. 

Strengths:

    • The authors used a good methodology to study the problem of pharmacy location in Wuhan. 

    • The data analysis and processing were both well implemented. 

Weaknesses:

    • In general terms the proposal is interesting. However, the authors may consider a generalization of the method for other studies in other regions. Also, this methodology could be used to deal with other location elections to retail. Thus, a general description of the method is required before publication. 

Round 2

Reviewer 2 Report

Basically, authors didn't perform the required experiments, mentioning the constraints in accessing the data necessary to make them.

Considering that they added some discussion concerning automation, and the focus of the journal, the paper can be accepted for publication.